# BRG1 governs glucocorticoid receptor interactions with chromatin and pioneer factors across the genome

Jackson A Hoffman[1†], Kevin W Trotter[1†], James M Ward[2], Trevor K Archer[1]*

[1]Epigenetics and Stem Cell Biology Laboratory, National Institute of Environmental Health Sciences, National Institutes of Health, North Carolina, United States; [2]Integrative Bioinformatics, National Institute of Environmental Health Sciences, National Institutes of Health, North Carolina, United States

**Abstract** The Glucocorticoid Receptor (GR) alters transcriptional activity in response to hormones by interacting with chromatin at GR binding sites (GBSs) throughout the genome. Our work in human breast cancer cells identifies three classes of GBSs with distinct epigenetic characteristics and reveals that BRG1 interacts with GBSs prior to hormone exposure. The GBSs pre-occupied by BRG1 are more accessible and transcriptionally active than other GBSs. BRG1 is required for a proper and robust transcriptional hormone response and knockdown of BRG1 blocks recruitment of the pioneer factors FOXA1 and GATA3 to GBSs. Finally, GR interaction with FOXA1 and GATA3 binding sites was restricted to sites pre-bound by BRG1. These findings demonstrate that BRG1 establishes specialized chromatin environments that define multiple classes of GBS. This in turn predicts that GR and other transcriptional activators function via multiple distinct chromatin-based mechanisms to modulate the transcriptional response.

DOI: https://doi.org/10.7554/eLife.35073.001

*For correspondence:
archer1@niehs.nih.gov

†These authors contributed equally to this work

Competing interests: The authors declare that no competing interests exist.

## Introduction

The Glucocorticoid Receptor (GR, encoded by the NR3C1 gene) is a type I nuclear receptor that elicits the transcriptional response to glucocorticoid steroid hormones. This transcriptional response is essential for human health and development. Glucocorticoids such the synthetic hormone Dexamethasone (Dex) are utilized to activate GR signaling to treat human auto-immune and inflammatory diseases and to promote fetal lung development. Understanding the mechanisms through which the transcriptional response to glucocorticoids is generated is critical for human health and for the further development of disease treatments. A detailed mechanism for GR transcriptional activity has been established through the examination of GR at model genes such as the Mouse Mammary Tumor Virus (MMTV). Upon hormone binding, GR enters the nucleus and binds to regions in the chromatin known as GR binding sites (GBSs). At MMTV, GR binding triggers the recruitment of other factors including the SWI/SNF chromatin remodeling complex (*Cordingley et al., 1987*; *Fryer and Archer, 1998*). Recruitment of the SWI/SNF complex induces the reorganization of nucleosomes around GR binding sites, which in turn facilitates binding of other transcription factors and potentiates transcriptional activation (*Archer et al., 1994*; *Wallberg et al., 2000*).

The mammalian SWI/SNF chromatin remodeling complex is comprised of one of two catalytic ATPases, BRG1 and BRM, and 10 or more BRM/BRG1-asssociated factor (BAF) subunits. The so-called BAF complex is critical throughout embryonic development and is among the most commonly mutated protein complexes in human cancers (*Shain and Pollack, 2013*; *Kadoch et al., 2013*; *Wu et al., 2017*). BRG1 and the BAF subunits also play critical roles in mediating the transcriptional response to glucocorticoid signaling. GR interacts directly with BAF57, BAF60A, and BAF250, and

**eLife digest** Steroid hormones play a number of roles in the body, including controlling the immune system and the body's response to stress. Artificially produced steroid hormones may also be used as part of treatments for cancer. The hormones affect the behavior of cells by binding to and activating hormone receptor proteins. The receptors can then interact with the cell's DNA to change the activity of nearby genes.

Gaining access to particular sites on a strand of DNA is not always easy. Cells pack DNA into a structure called chromatin. In some regions the DNA is so tightly wrapped in the chromatin that the receptors cannot access it. The structure of the chromatin therefore affects how a cell responds to steroid hormones.

Inaccessible regions of chromatin can be 'opened up' by two groups of proteins, known as remodeling proteins and pioneer factors. Hormone receptors can work with these proteins to access particular DNA regions, but exactly how all these proteins work together was not fully understood.

Hoffman et al. have now used DNA and RNA sequencing technologies to examine the roles of a hormone receptor called the glucocorticoid receptor, a remodeling protein called BRG1, and various pioneer factors in human breast cancer cells. This revealed three ways in which the glucocorticoid receptors worked with the other proteins when binding to chromatin. These could be distinguished by the pattern of BRG1 molecules bound to the DNA.

Further investigation showed that BRG1 controls how the glucocorticoid receptor affects the activity of genes. In addition, BRG1 influences how the receptor interacts with pioneer factors when it is bound to DNA. Future research into how these proteins work together could ultimately help us to improve how we use steroid hormones to treat diseases.

DOI: https://doi.org/10.7554/eLife.35073.002

requires the catalytic ATPase activity of BRG1 to promote transcriptional activation of MMTV (*Nie et al., 2000*; *Inoue et al., 2002*; *Hsiao et al., 2003*). Chromatin remodeling by the BAF complex was required for the subsequent recruitment of RNA Polymerase II and other transcription factors to the MMTV promoter (*Johnson et al., 2008*). Transcriptional activation of MMTV was also promoted by the recruitment of a complex containing Ku70/86, Topoisomerase IIβ, and Poly(ADP-ribose) polymerase one by BRG1 (*Trotter et al., 2015*). Thus, chromatin remodeling through the BAF complex is a critical component of GR signaling.

Beyond the requirement for chromatin remodeling by the BAF complex, the underlying chromatin landscape appears to play a crucial role in patterning the hormone response. GR preferentially binds to regions in the chromatin that are pre-accessible as measured by DNase hypersensitivity or formaldehyde-assisted isolation of regulatory elements (*John et al., 2011*; *Burd et al., 2012*). These findings indicated that GR chromatin interactions were predetermined by other chromatin interacting factors. Pioneer factors, transcription factors that can bind to and open regions of closed chromatin, have been implicated in the pre-patterning of GR binding. The pioneer factors C/EBPβ and AP1 pre-occupied a large proportion of GR binding sites in mouse liver and mammary cells, and were required to maintain chromatin accessibility at GR binding sites (*Biddie et al., 2011*; *Grøntved et al., 2013*). Similarly, FOXA1 pre-bound a large number of Estrogen Receptor (ER, another nuclear hormone receptor closely related to GR) binding sites and was required for ER binding and transcriptional activity. Conversely, recent work has demonstrated that hormone signaling through both ER and GR promoted the redistribution of FOXA1 chromatin interactions (*Swinstead et al., 2016*). These findings helped to demonstrate that current models of GR activity fail to fully account for the complexities of GR signaling at a genomic scale, and that more sophisticated and diverse models are required to describe the mechanisms through which GR initiates a transcriptional response.

In this study, we examined mechanisms of GR transcriptional regulation through genome-scale analyses of hormone-induced changes in transcriptional activity and the binding patterns of GR, BRG1, and pioneer factors. We identify distinct classes of GR binding site based upon the binding profile of BRG1 before and after hormone treatment. BRG1 binding to GR sites prior to hormone marked GR binding sites that were pre-accessible and enriched for marks of transcriptionally active

chromatin. BRG1 was required for a robust GR transcriptional response, as disruption of BRG1 expression dramatically altered the profile of hormone-induced differentially expressed genes. GR binding sites that were pre-bound by BRG1 were also enriched for motifs of pioneer factors such as FOXA1 and GATA3, and BRG1 binding at pioneer factor binding sites in untreated cells was predictive of GR binding upon hormone treatment. Furthermore, BRG1 expression was required for Dex-induced recruitment of additional FOXA1 and GATA3 to GR binding sites. Taken together, our data suggest that GR elicits the transcriptional response to hormone via multiple distinct mechanisms that are reliant on the pre-patterning of specialized chromatin environments through the actions of the BAF complex and additional factors.

## Results

### Differential patterns of BRG1 interaction define three classes of GR binding site

Current models of GR function commonly depict the hormone-dependent recruitment of the BRG1-containing SWI/SNF chromatin remodeling complex to GBSs (*Cordingley et al., 1987*; *Fryer and Archer, 1998*; *Archer et al., 1994*; *Wallberg et al., 2000*). This recruitment of BRG1 facilitates the opening of chromatin around the GBS to enhance the ability of GR to elicit transcriptional effects. However, recent work demonstrates that GBSs exhibit chromatin accessibility prior to hormone treatment, suggesting that some mechanism pre-patterns the chromatin environment around GBSs (*John et al., 2011*; *Burd et al., 2012*).

To investigate a potential role for BRG1 in this phenomenon, we preformed chromatin immunoprecipitation with high-throughput sequencing (ChIP-seq) in the A1-2 model cell line (*Figure 1*) (*Archer et al., 1994*). We obtained data of high depth (>60 million reads per GR or BRG1 ChIP-seq) and called peaks using a 0.001 false discovery rate cutoff to ensure high confidence in identifying GR and BRG1 binding sites. One hour of hormone exposure was sufficient to induce a massive DNA binding response by GR, with 29934 GR binding sites/peaks identified specifically in Dex-treated cells (*Figure 1A*). The number of peaks called from our dataset falls within the range of peak numbers called by GR ChIP-seq experiments in other cell lines (*Swinstead et al., 2016*; *Starick et al., 2015*). Dex treatment also had a robust effect on the chromatin localization of BRG1. While 50,000 + BRG1 peaks were identified in each condition, only 33582 peaks were shared while 17699 were specific to EtOH-treated cells and 20658 were specific to Dex-treated cells (*Figure 1C*, *Figure 1—figure supplement 1A*). This rearrangement of BRG1 chromatin localization is consistent with BRG1 being recruited to GBSs upon hormone treatment. To verify this, we examined the overlap between BRG1 and GR peaks and found that 58% of GR peaks are overlapped by a BRG1 peak (*Figure 1B,C*). Surprisingly, there was significant overlap in both EtOH and Dex-treated cells (*Figure 1B*), with 12034 GR peaks bound by BRG1 in both conditions, and 5565 bound by BRG1 in a Dex-specific manner (*Figure 1C*). 12223 GR peaks were not bound by BRG1, and a total of 54340 BRG1 peaks (including 15093 Dex-specific and 17699 EtOH-specific) did not overlap GR (*Figure 1C*, *Figure 1—figure supplement 1B*). These findings indicate that a large subset of the subsequent GR peaks are bound by BRG1 prior to hormone treatment consistent with the concept that BRG1 could be involved in pre-patterning GBSs.

To further dissect the relationship between GR and BRG1, we defined three classes of GR peak: Class I peaks as GR peaks lacking any overlap with a BRG1 peak, Class II peaks as GR peaks overlapped by BRG1 peaks in both EtOH- and Dex-treated conditions, and Class III peaks as GR peaks that overlapped only Dex-specific BRG1 peaks (*Figure 1C*). Collectively, Class I GR peaks were narrower and showed less overall GR enrichment than Class II or Class III peaks (*Figure 1D,F*). BRG1 was not enriched at Class I GR peaks, was constitutively enriched at Class II GR peaks, and was induced by Dex-treatment at Class III peaks (*Figure 1E,G*). The peak classes were easily identifiable at gene level coverage (*Figure 1H–J*) and GR and BRG1 enrichment patterns were independently validated by ChIP-QPCR (*Figure 1K*). Thus, we utilized these three GR peak classes in our subsequent analyses to examine how differential patterns of BRG1 interaction could define the GR-mediated hormone response.

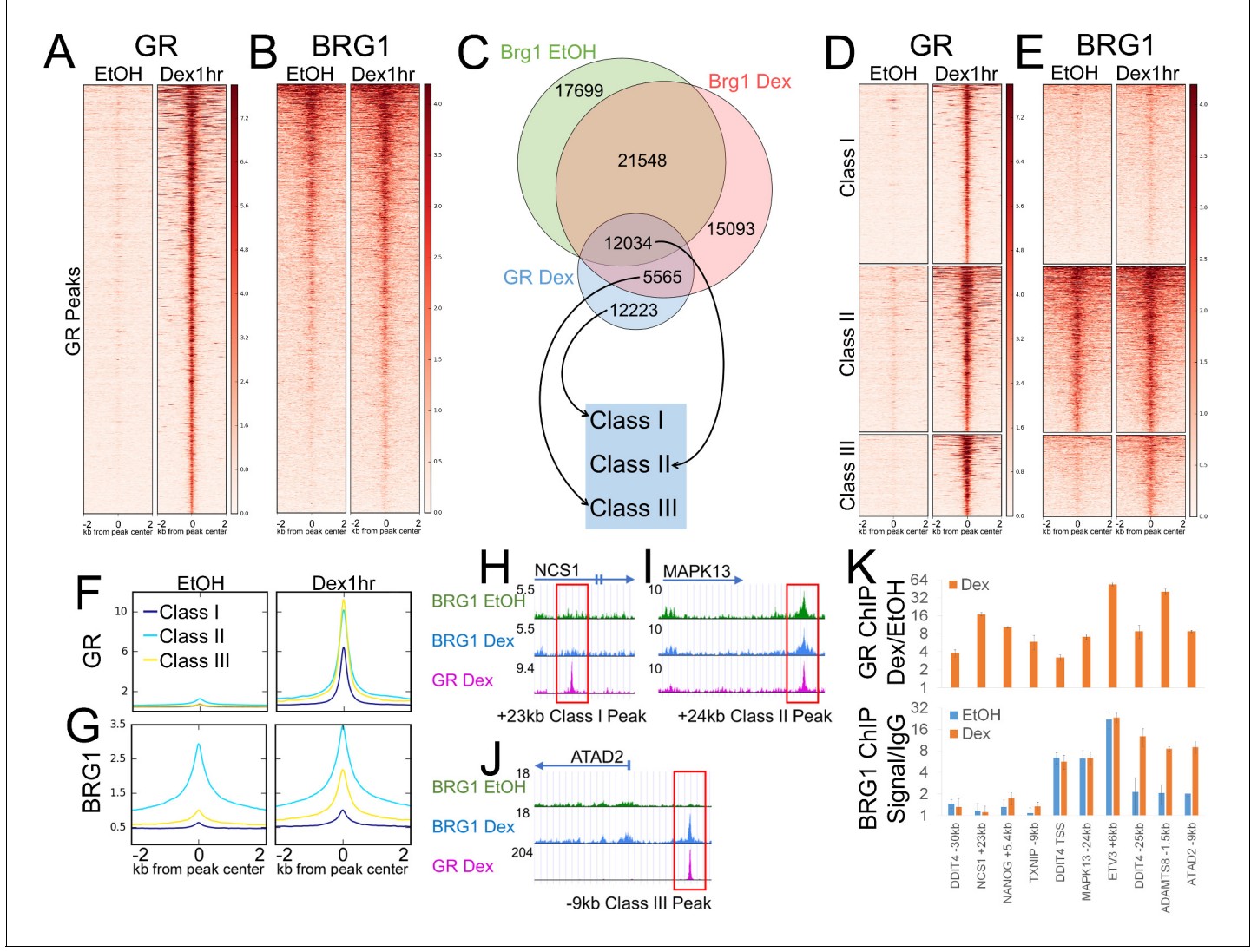

**Figure 1.** BRG1 chromatin interaction defines distinct classes of GR binding site. (**A**) Heatmap demonstrating that GR is detected at 29934 binding sites following 1 hr Dex treatment. (**B**) Heatmaps illustrating detection of BRG1 at a subset of GR binding sites prior to hormone treatment and recruitment of BRG1 to additional sites following 1 hr of Dex treatment. (**C**) Venn diagram of overlap between GR, BRG1-EtOH, and BRG1-Dex peaks and the designation of three classes of GR peaks. (**D**) Heatmap of GR signal over GR peaks divided into three classes. (**E**) Heatmap of BRG1 signal over GR peaks divided into three classes. (**F–G**) Meta-profiles of GR and BRG1 ChIP-seq coverage over GR peak classes. (**H–J**) UCSC genome browser snapshots of GR, BRG1 EtOH, and BRG1 Dex ChIP-seq coverage at representative Class I, II, and III GR peaks. (**K**) ChIP-QPCR validation of GR and BRG1 ChIP enrichment at representative Class I, II, and III GR peaks.

DOI: https://doi.org/10.7554/eLife.35073.003

The following figure supplements are available for figure 1:

**Figure supplement 1.** Hormone treatment reorganizes BRG1 chromatin localization.
DOI: https://doi.org/10.7554/eLife.35073.004

**Figure supplement 2.** Biological replication of Next Generation Sequencing data sets.
DOI: https://doi.org/10.7554/eLife.35073.005

## Class II GR peaks are associated with open and transcriptionally active chromatin

Given the differences in BRG1 distribution we next sought to determine whether the GR peak classes also exhibited distinct chromatin environments. Short, sub-nucleosome length ATAC-seq reads were used as a measure of chromatin accessibility, and were strongly enriched at Class II peaks independent of Dex treatment. Thus, Class II GR peaks are accessible prior to hormone treatment

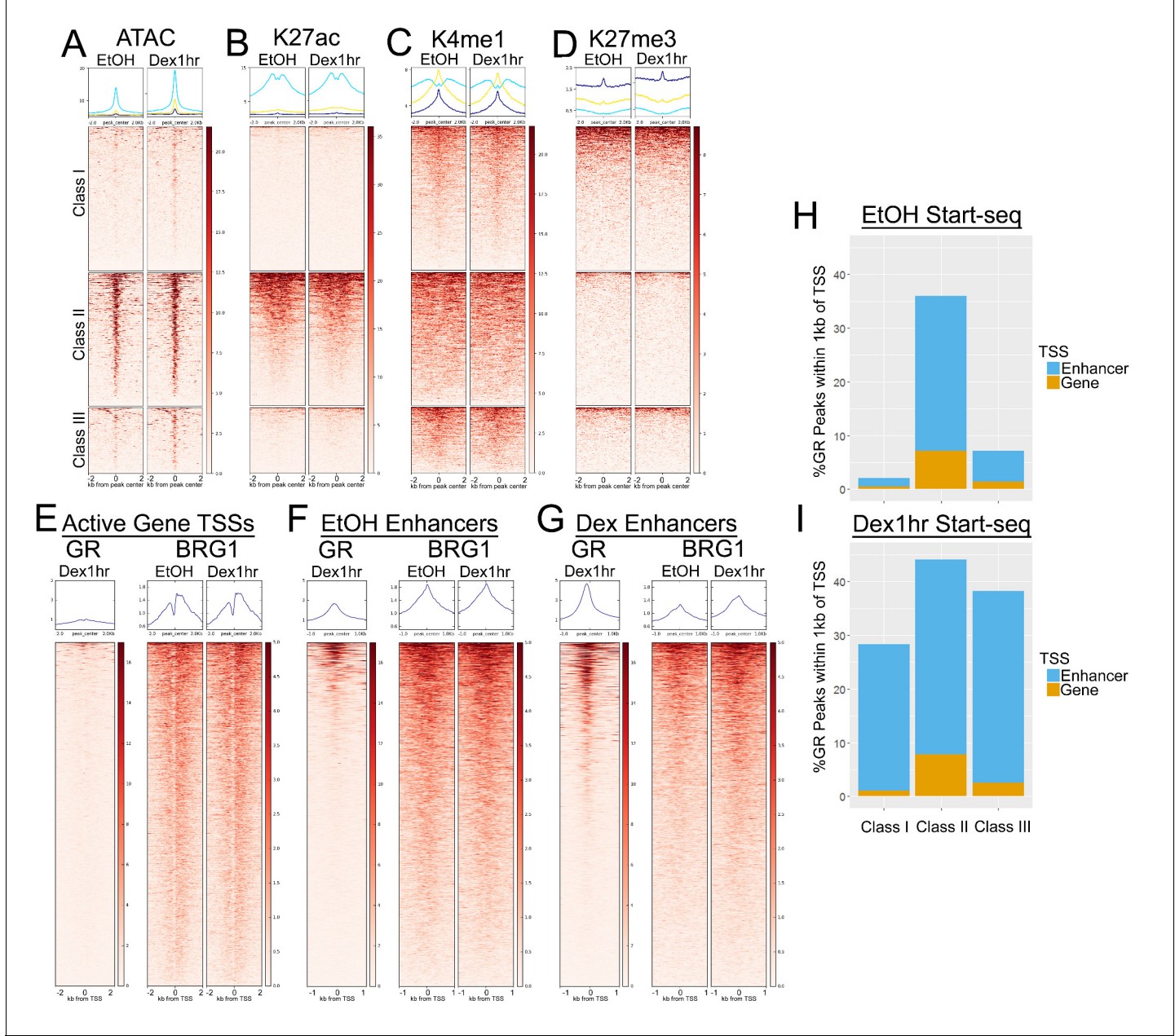

**Figure 2.** Class II GR peaks are associated with open and transcriptionally active chromatin. (**A**) ATAC-seq accessibility/short reads are enriched over Class II GR peaks independently of hormone treatment. (**B**) H3K27ac ChIP-seq coverage is specifically enriched over Class II GR peaks independently of hormone treatment. (**C**) H3K4me1 ChIP-seq is enriched over Class I and III peaks, whereas Class II peaks have a broader enrichment pattern with a trough directly over the GR peak. (**D**) H3K27me3 is not strongly enriched at any of the GR peak classes. (**E–G**) Heatmaps depicting GR and BRG1 ChIP-seq signal around active gene TSSs (**E**), enhancer TSSs from EtOH/untreated cells (**F**), and enhancer TSSs from Dex 1 hr cells (**G**). (**H–I**) Stacked barplots showing the percentage of GR peaks that are within 1 kb of an active gene TSSs (orange) or enhancer TSSs (blue) in using Start-seq TSS calls from untreated cells (**H**) and Dex 1 hr cells (**I**).

DOI: https://doi.org/10.7554/eLife.35073.006

The following figure supplements are available for figure 2:

**Figure supplement 1.** BRG1 peaks without GR are accessible and have active chromatin marks.
DOI: https://doi.org/10.7554/eLife.35073.007
**Figure supplement 2.** GR is weakly enriched around TSSs.
DOI: https://doi.org/10.7554/eLife.35073.008
**Figure supplement 3.** BRG1 is enriched around TSSs.

*Figure 2 continued on next page*

*Figure 2 continued*

DOI: https://doi.org/10.7554/eLife.35073.009

(*Figure 2A*). Conversely, minimal accessibility was detected at Class I and III GR peaks in EtOH-treated cells, indicating that prior to hormone treatment, these GR peaks were largely inaccessible (*Figure 2A*). Upon 1 hr Dex exposure, low but distinct levels of accessibility were induced, predominantly at class III peaks where BRG1 binding was also induced (*Figure 2A*). Thus, chromatin accessibility at GR peak classes was directly correlated with the pattern of BRG1 occupancy.

We next examined histone modifications at the GR peak classes for differentially enriched active or repressive marks. Histone 3 Lysine 27 acetylation (K27ac), a histone modification associated with transcriptionally active chromatin at TSSs and active enhancers, was only detected at Class II GR peaks (*Figure 2B*). Histone 3 Lysine four monomethylation (K4me1), a marker of both inactive/poised and active enhancers, was enriched at all three peaks classes, but had a unique pattern at Class II peaks (*Figure 2C*). While K4me1 enrichment was centered on the GR peak at Class I and III peaks, Class II peaks had broader K4me1 enrichment with an apparent trough of enrichment directly over the GR peak (*Figure 2C*). None of the peak Classes displayed strong enrichment of Histone 3 Lysine 27 trimethylation (K27me3), a repressive chromatin modification (*Figure 2D*). Taken together with the ATAC-seq data, the patterns of histone modifications at GR peaks were associated with enhancer-like chromatin marks. The strong ATAC accessibility signal and K27ac enrichment at Class II peaks suggested that they might represent GR binding to active enhancers. On the other hand, the relatively low levels of ATAC accessibility and the patterns of K4me1 and K27ac at Class I and III peaks suggested that they represented GR binding to inactive or poised enhancers. Furthermore, Class II GR peaks represent a distinct set of GR peaks that are associated with BRG1 as well as marks of accessible and transcriptionally active chromatin.

To characterize transcriptional events associated with GR chromatin interaction, we investigated whether any GR peaks were proximal to functionally engaged transcriptional start sites (TSSs). We previously used Start-seq to identify actively transcribed TSSs in A1-2 cells (*Lavender et al., 2016*). We divided these TSSs into those correlated with active, annotated gene TSSs and those that were greater than 2 kb from any gene TSS and represent putative active enhancer TSSs. GR ChIP-seq signal after 1 hr of Dex treatment was modestly enriched over active gene TSSs (*Figure 2—figure supplement 2*), however this was dwarfed by the level of GR enrichment at active enhancer TSSs identified in either EtOH- and Dex-treated cells (*Figure 2E–G*). The average GR enrichment over active gene TSSs was not markedly increased when the analysis was restricted to genes that were differentially expressed (DEGs) following 1, 4, 8, or 18 hr of Dex treatment (*Figure 2—figure supplement 2*) Thus, GR was much more frequently associated with enhancer transcription than gene transcription. Unlike GR, BRG1 ChIP-seq signal was broadly detected and similarly enriched at all active TSSs (*Figure 2E*). Over active gene TSSs and EtOH-detected active enhancers, the average levels BRG1 enrichment appeared unaffected by Dex (*Figure 2E–F*). This was consistent with the predominantly Class II-specific enrichment of ATAC-seq and H3K27ac ChIP-seq signal and indicated that BRG1 was constitutively associated with most active TSSs in A1-2 cells. Furthermore, ATAC-seq accessibility and K27ac were also strongly enriched at hormone-independent BRG1 peaks that did not overlap GR peaks (*Figure 2—figure supplement 1*). Taken together, these data indicated that BRG1 was largely associated with open and active chromatin independently of GR, and that BRG1 peaks that were not affected by hormone treatment most strongly exhibited these characteristics. However, a Dex-induced increase in BRG1 enrichment was observed at active enhancer TSSs detected in Dex-treated cells (*Figure 2G*), and more modestly at the TSSs of Dex-induced DEGs (*Figure 2—figure supplement 3*). Thus, TSSs with altered transcriptional activity upon Dex exposure also exhibited hormone-induced BRG1 enrichment.

K27ac and K4me1 ChIP-seq suggested that GR peak classes were differentially associated with active and inactive/poised enhancers (*Figure 2B–C*). To further dissect this relationship, we looked to see how many peaks in each class were in close proximity to active enhancer and gene TSSs. When considering active TSSs called in untreated cells, 38% of Class II peaks were within 1 kb of a TSS, compared to 2% of Class I peaks and 7% of Class III peaks (*Figure 2H*). However, when considering active TSSs called in cells treated with Dex for 1 hr, a significant portion of each peak class was

within 1 kb of a TSS (26% Class I, 45% Class II, and 38% Class III) (*Figure 2I*). Despite the induction of transcriptional activity near a significant subset of Class I and III peaks, K27ac was not observably induced at these peaks (*Figure 2B*) whereas the pattern of K4me1 was unaffected (*Figure 2C*). Taken together with the patterns of K27ac and K4me1, these data indicated that Class II peaks represented GR interactions with accessible chromatin and transcriptionally active enhancers. Furthermore, they indicated that Class I and III peaks represented GR interactions with inactive or poised enhancers that could be activated upon GR binding, but exhibited limited accessibility and were devoid of the K27ac mark.

## BRG1 is required for the transcriptional hormone response

To further investigate the role of BRG1 in regulating the transcriptional hormone response, we performed RNA-seq in cells which harbor an inducible shRNA targeting BRG1 (A1A3 cells, previously described in [*Burd et al., 2012*]). Treatment with Doxycycline for 72 hr resulted in an 80–85% reduction in BRG1 protein levels as well as partial reduction in the nuclear levels of GR protein (*Figure 3—figure supplement 1A*). RNA-seq performed at 1 hr Dex treatment in A1-2 cells yielded approximately 200 DEGs (*Lavender et al., 2016*). In order to capture a more robust transcriptional hormone response for analysis, we used an 8 hr Dex treatment in A1A3 cells. In normal conditions, 1244 DEGs (Fold Change > 1.5, p-value<0.01, false discovery rate <0.05) were identified following 8 hr of Dex treatment (*Figure 3A*). 743 of these DEGs failed to meet the same fold-change and significance cutoffs in Dex treated BRG1-KD cells, indicating BRG1 was required the transcriptional response to Dex. Intriguingly, BRG1-KD cells had 114 Dex-regulated DEGs that were not called DEGs in control Dex-treated cells, indicating that BRG1 also suppressed the hormone responsiveness of a small number of genes (*Figure 3A*). Visualizing the changes in gene expression by heatmap revealed that while the hormone response was largely muted or suppressed following BRG1 knockdown, a significant number of genes showed equal or greater hormone responses following BRG1 knockdown (*Figure 3B*). Indeed, the absolute fold change of both 'common' and 'lost' DEGs was reduced in BRG1-KD cells, and increased in 'gained' DEGs (*Figure 3C*). Together, these data indicated that BRG1 was required for both a robust transcription hormone response and to suppress ectopic hormone responsiveness.

Decreased levels of GR protein in BRG1-KD cells suggested that part of the BRG1 effect might instead result from insufficient nuclear GR. While the increased hormone-responsiveness of the 114 'gained' DEGs in BRG1-KD cells served as evidence that this was not the case, we sought to directly test whether decreased GR was the dominant factor in the loss of dex-induction of 'lost' DEGs. Reduction of GR protein levels by 50–70% by siRNA resulted in a modest loss of dex-induced transcription at candidate 'lost' genes (*Figure 3—figure supplement 1B–C*). However, BRG1-KD resulted in a much stronger loss of dex-induced transcription (*Figure 3—figure supplement 1C*). As such, these data strongly suggest that the changes in the Dex response in A1A3 cells are predominantly driven by the silencing of BRG1, and not by the more modest decrease in nuclear GR levels.

We sought to correlate gene expression changes in A1A3 cells with the presence and proximity of GR peaks. DEG TSSs tended to be closer to GR peaks, and the percentage of DEGs with GR peaks within 50 kb was more than double that of non-DEG obsTSSs (77.2 to 29.8%). Thus, while GR binding is distal from gene TSSs, genes that are regulated by GR tend to have a higher degree of local GR binding sites. When considering the closest GR peak to each DEG, the different types of DEGs had different proportions of GR peaks classes, with Class II peaks enriched among the closest peaks to 'common' and 'gained' DEGs (*Figure 3D*). Comparing the distance from the closest GR peak to each DEG TSS, Class II peaks were also closer than Class I or Class III peaks (*Figure 3E*). Overall, 'common' DEGs had closer GR peaks than 'gained' or 'lost' DEGs, with the median distance from TSSs to closest GR peaks of 6889, 20094, and 22104 bp, respectively (*Figure 3F*). DEGs that were 'gained' or 'lost' also had fewer GR peaks within 50 kb of their TSSs (*Figure 3G*). Taken together, these data indicate that BRG1 presence was more critical for hormone responsiveness at genes where GR binding was the most distal.

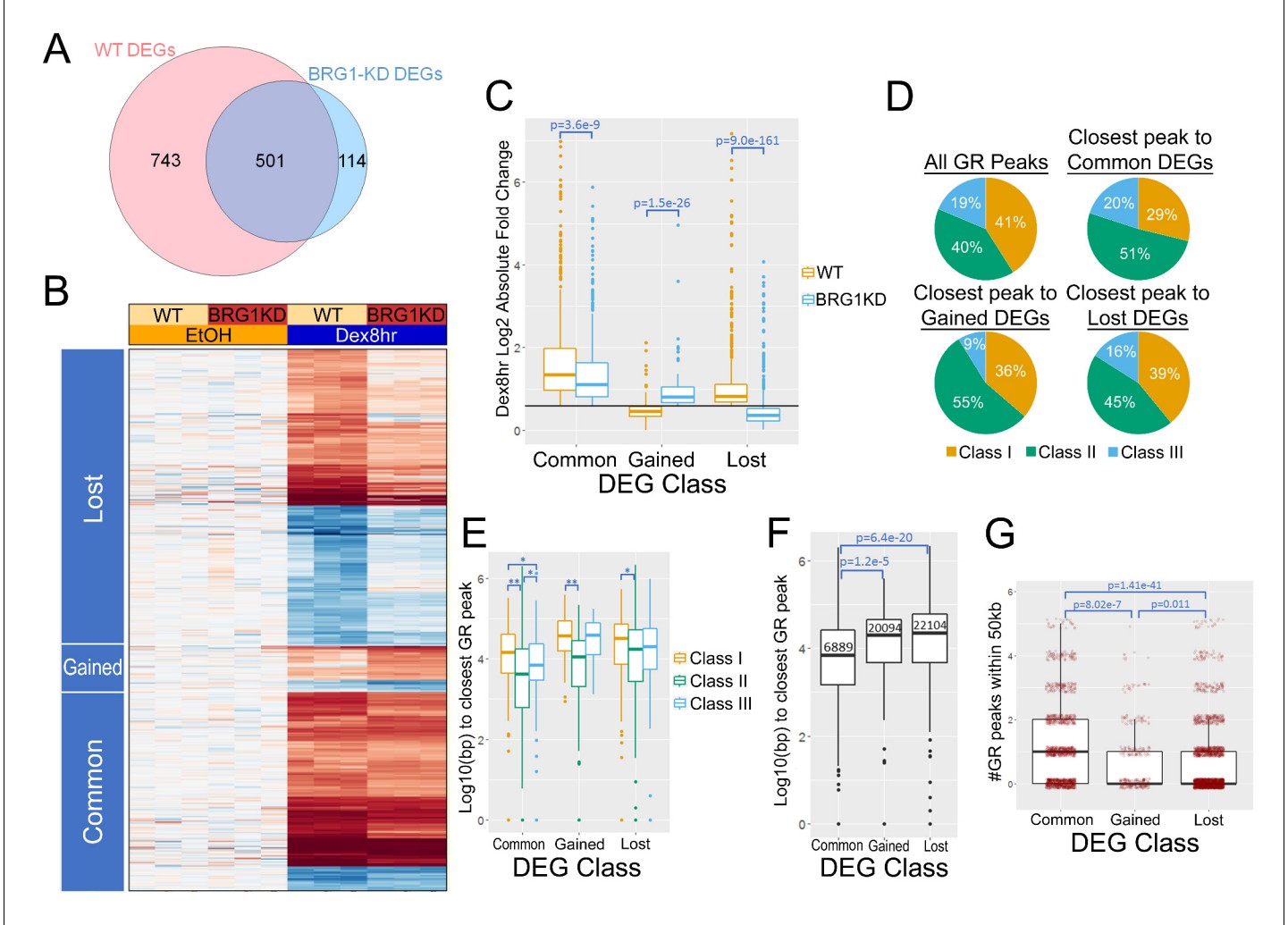

**Figure 3.** BRG1 is required for the Dex-induced transcriptional response. (A) Venn diagram showing overlap between DEGs in 8 hr Dex treatment vs EtOH in WT and BRG1-KD cells. (B) Heatmap depicting log2 fold change of DEGs in 8 hr Dex treatment vs EtOH in WT and BRG1-KD cells. (C) Box plots of log two absolute fold changes of 8 hr Dex DEGs. Black line depicts 1.5 fold change. (D) Pie charts depicting the class of the closest GR peak to each DEG TSS. (E) Box plots depicting the log 10 distance from the DEG TSSs to their closest GR peaks divided into GR peak classes. *=p value<0.05, **=p value<0.001 (F) Box plots showing the distance from DEG TSSs to their closest GR peaks. Median distances are labeled in base pairs. (G) Box and scatter plots depicting the number of GR peaks within 50 kb of the TSS of 8 hr Dex DEGs. Outlier peaks with more than 5 GR peaks within 50 kb of the TSS were omitted from the graph for display purposes. All p-values were calculated using Wilcoxon rank sum tests.

DOI: https://doi.org/10.7554/eLife.35073.010

The following figure supplement is available for figure 3:

**Figure supplement 1.** BRG1 and GR knockdown in A1A3 cells.

DOI: https://doi.org/10.7554/eLife.35073.011

## GR peak classes have distinct underlying DNA sequences and transcription factor motifs

GR frequently binds to degenerate GR recognition sequences or GR response elements (GREs) (*Starick et al., 2015*) and can also interact with other regions of the genome through cooperation with or tethering by other transcription factors such as AP-1, NFκB, and STAT proteins (*Biddie et al., 2011*; *Rao et al., 2011*; *Langlais et al., 2012*). To determine whether our GR peaks classes segregated distinct sequence specificities, we first searched under GR peaks for perfect GREs. Using the total set of GR peaks, perfect GREs were found under 28.3% of the peaks and GREs with single mismatches were found under an additional 51% of peaks (*Figure 4A*). Perfect

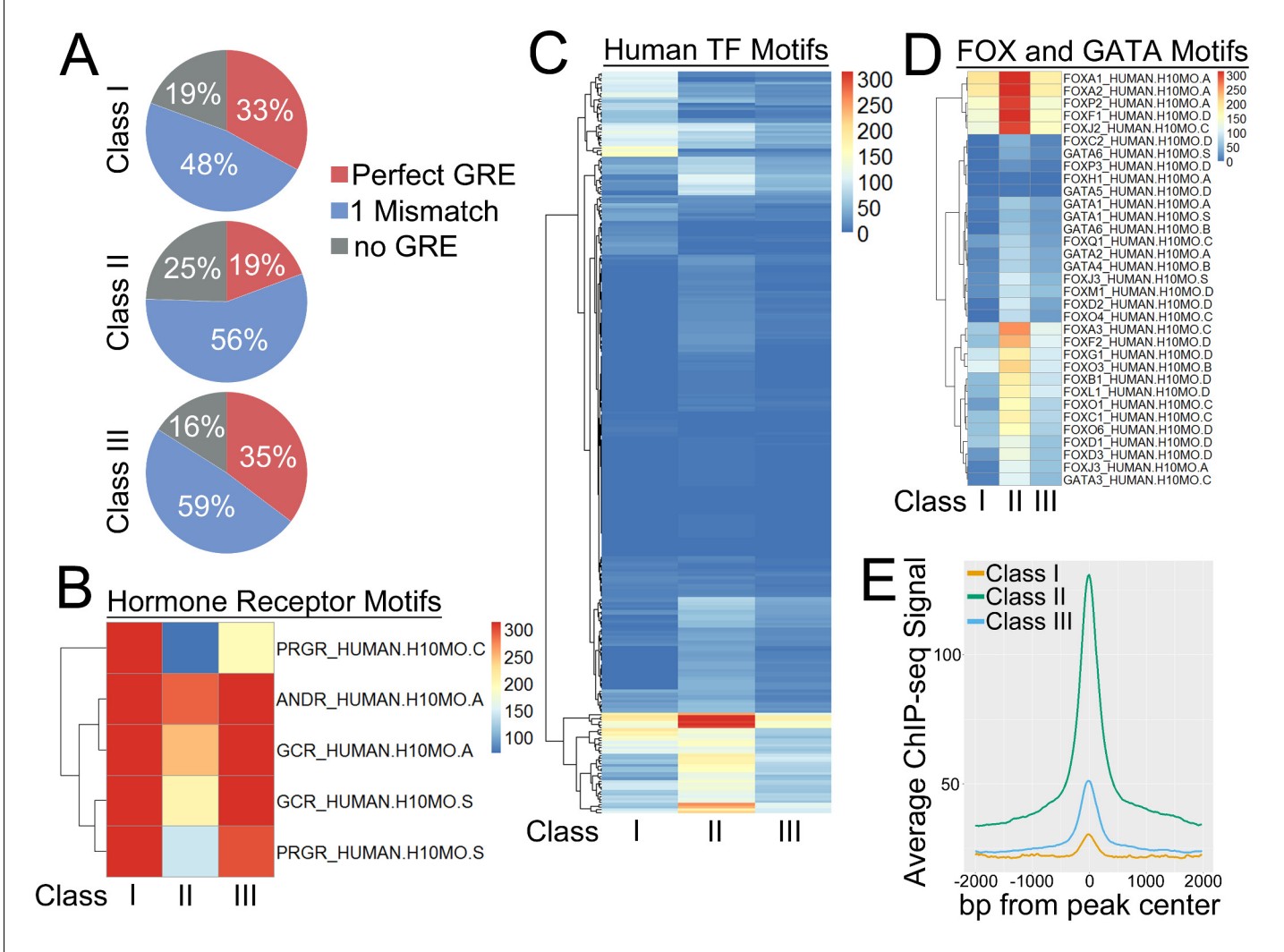

**Figure 4.** GR peak classes have distinct underlying DNA sequences and transcription factor motifs. (A) Class II GR peaks have fewer perfect GRE motifs than Class I or Class III peaks. (B) Heatmap of -log10 p-values for enrichment of hormone receptor motifs under GR peak classes. (C) Heatmap of -log10 p-values for enrichment of human transcription factor motifs under GR peak classes. (D) Heatmap of -log10 p-values for enrichment of FOX and GATA motifs under GR peak classes. (E) Meta-profile of average coverage of 25 ENCODE transcription factor ChIP-seq MCF7 datasets over GR peak classes.

DOI: https://doi.org/10.7554/eLife.35073.012

The following source data and figure supplement are available for figure 4:

**Source data 1.** Table of MCF7 ENCODE ChIP-seq data sets.

DOI: https://doi.org/10.7554/eLife.35073.014

**Figure supplement 1.** Transcription factor motifs enriched under distinct GR peak classes.

DOI: https://doi.org/10.7554/eLife.35073.013

GREs were more common in Class I and Class III peaks (33 and 35.3%, respectively) than Class II peaks (19.4%) (*Figure 4A*). Motif analysis revealed similar patterns, with GREs, Androgen Receptor motifs, and Progesterone Receptor motifs being strongly enriched under all three peak classes, but with the lowest enrichment levels under Class II peaks (*Figure 4B*).

Conversely, Class II peaks were most strongly enriched for other transcription factors. Motif analysis revealed that Class II peaks exhibited the highest average level of motif enrichment (*Figure 4C*). This was largely driven by FOX and GATA motifs, which were more strongly enriched under Class II peaks than Class I or Class III peaks (*Figure 4D*). To validate these predictive analyses, we pulled data from 25 ENCODE transcription factor ChIP-seq experiments in Mcf7 cells (*Figure 4—source data 1*) and generated a meta-profile of transcription factor ChIP enrichment over the three GR

peak classes (*Figure 4E*). Class II GR peaks displayed the strongest levels of enrichment, while Class I and Class III peaks showed only low to moderate levels of enrichment (*Figure 4E*). Motif analyses yielded several other interesting motif families to consider in the context of BRG1 and the GR peak classes. SP/KLF and POU motifs were specifically enriched under Class II and Class III peaks (*Figure 4—figure supplement 1*), which suggested that GR binding in cooperation with these transcription factor families may also involve BRG1. On the other hand, STAT, NFATC, and OLIG motifs were most strongly enriched under Class I GR peaks (*Figure 4—figure supplement 1*), indicative of transcription factor interactions that may occur in the absence of BRG1. Taken together, these analyses revealed that the three GR peak classes had distinguishable DNA sequence content and that the BRG1-GR interaction could be moderated by other transcription factors.

## BRG1 is required for Dex-induced recruitment of pioneer factors to GR binding sites

Recent work has suggested that the hormone response is coordinated by functional interactions between nuclear hormone receptors and pioneer factors such as FOXA1 and GATA3 (*Biddie et al., 2011*; *Grøntved et al., 2013*; *Carroll et al., 2005*; *Holmqvist et al., 2005*; *Laganiere et al., 2005*; *Hurtado et al., 2011*). As we observed differential enrichment of FOXA1 and GATA3 motifs under the GR peak classes, we performed ChIP-seq to examine the interaction of these factors at each peak class. Both FOXA1 and GATA3 showed strong levels of enrichment at Class II GR peaks in both untreated and 1 hr Dex treated cells (*Figure 5A–B*). Class I and Class III peaks had similarly low levels of FOXA1 and GATA3 in untreated cells (*Figure 5A–B*). However, at Class III peaks, there was a marked increase in the detected levels of FOXA1 and GATA3 binding upon 1 hr of Dex treatment (*Figure 5A–B*) comparable to the Dex-induced enrichment of BRG1 at these peaks

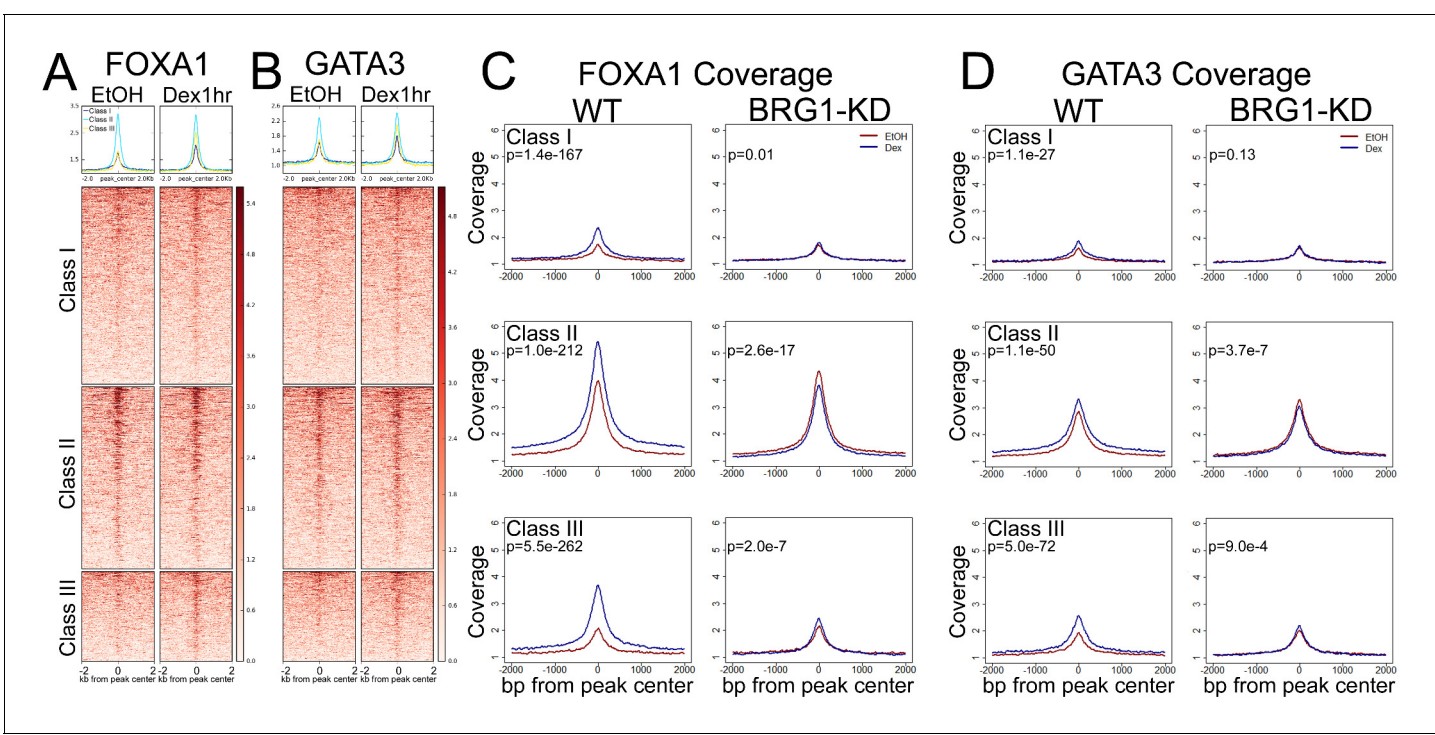

**Figure 5.** BRG1 is required for Dex-induced recruitment of pioneer factors. (**A–B**) FOXA1 and GATA3 ChIP-seq coverage is enriched across all GR peak classes, but shows strongest enrichment over Class II peaks. (**C–D**) Meta-profiles of FOXA1 and GATA3 ChIP-seq signal over GR peak classes in A1A3 cells. In wild-type cells, Dex treatment induced recruitment of additional FOXA1 and GATA3 to all three GR peak classes. This recruitment is lost following BRG1-KD. To calculate P-values in (**C**) and (**D**), we used the average signal in the 500 bp window centered on the center of the GR peak, and performed unpaired Wilcoxon Rank Sum/Mann-Whitney test.

DOI: https://doi.org/10.7554/eLife.35073.015

(*Figure 1E*). Thus, at GR peaks, the pattern of pioneer factor binding correlated with that of BRG1 binding.

To determine whether BRG1 was required for pioneer factor binding at GR peaks, we performed GR and FOXA1 ChIP-seq in A1A3 cells. The levels of FOXA1 and GATA3 protein were similar between control and BRG1-KD cells (*Figure 3—figure supplement 1*), indicating that BRG1 was not required for their expression. In control cells treated with Dex for 1 hr, a Dex-induced increase in FOXA1 and GATA3 enrichment was observed at all three GR peak classes, with the most substantial increase occurring at Class III peaks (*Figure 5C–D*, left columns)). In vehicle treated BRG1-KD cells, the loss of BRG1 appeared to have little effect on the enrichment of FOXA1 and GATA3 at GR peaks (*Figure 5C–D*, red lines). In contrast, the DEX-induced increase in FOXA1 and GATA3 enrichment at GR peaks was almost completely blocked in BRG1-KD cells (*Figure 5C–D*, blue lines). Taken together, these experiments demonstrate that BRG1 was not required for pioneer factor interaction at GR peaks. However, BRG1 was required for hormone-induced changes in pioneer factor enrichment at GR peaks.

## BRG1 binding to pioneer factor peaks is predictive of GR binding upon hormone treatment

We next sought to take a pioneer factor-centric approach to determine whether the presence of BRG1 affected the interaction of GR with pioneer factors. In vehicle-treated cells, BRG1 peaks intersected 16.4% of FOXA1 peaks (1594 peaks, *Figure 6A,D*) and 14.5% of GATA3 peaks (1145 peaks, *Figure 6B,E*). These peaks were predominantly unique, with only 249 of the FOXA1 +BRG1 peaks intersecting a GATA3 +BRG1 peak, similar to the overall proportion of overlap between FOXA1 and GATA3 peaks (*Figure 6C*). For both FOXA1 and GATA3, perfect GRE motifs were present at similar proportions between peaks with or without BRG1, with approximately 31% of FOXA1 peaks and 28% of GATA3 peaks having a perfect GRE motif within 500 bp of the center of the peak (*Figure 6—figure supplement 1*). Despite this, GR binding at FOXA1 and GATA3 peaks was almost entirely restricted to peaks that intersected BRG1 peaks (*Figure 6D–E*). As BRG1 was present at these sites in both untreated and Dex-treated cells (*Figure 6D–E*), these are Class II GR peaks. ATAC-seq nucleosome-free reads and K27ac ChIP-seq signal were also predominantly restricted to pioneer factor peaks that intersected BRG1 (*Figure 6D–E*). Thus, the presence of BRG1 at pioneer factor peaks in untreated cells was predictive of subsequent GR binding upon Dex treatment. These findings suggest that BRG1 is involved in pre-patterning a subset of pioneer factor binding sites to facilitate GR binding upon hormone treatment and that pioneer factor binding alone is not predictive of GR binding. Class II GR peaks represent GR binding to regions of chromatin that are pre-patterned by BRG1 and pioneer factors.

## Discussion

The requirement for BRG1-mediated chromatin remodeling in potentiating the transcriptional response to glucocorticoid signaling at model genes (e.g. MMTV) was established over two decades ago. Our examination of the genomic glucocorticoid response demonstrates a previously undescribed role of BRG1 in patterning the underlying chromatin architecture. Our data reveals that BRG1 interacts with approximately 40% of GR binding sites prior to hormone treatment, and an additional 20% of GR binding sites upon hormone treatment. BRG1 is also broadly associated with transcriptional activity at active gene and enhancer TSSs. The patterns of BRG1 binding at GR biding sites prior to and upon hormone signaling allowed us to define three classes of GR binding site (*Figure 7*). These classes exhibited distinct patterns of underlying chromatin accessibility, transcriptional activity, histone modification, and transcription factor motif enrichment and binding. These findings are corroborated by the observation that GR bound enhancers exist in three distinct chromatin states in mouse mammary adenocarcinoma cells (*Johnson et al., 2018*). Class I and Class III peaks gain chromatin accessibility upon Dex exposure and are associated with Dex-specific enhancer TSSs, suggesting that they represent regulatory elements that are activated only upon hormone treatment. Class II GR binding sites are strikingly enriched for chromatin that is active and accessible prior to hormone signaling and appear to represent GR binding to active enhancers.

Our examination of GR binding sites yielded several interesting observations regarding the nature of GR binding and transcriptional activity. The majority of GR binding sites are not

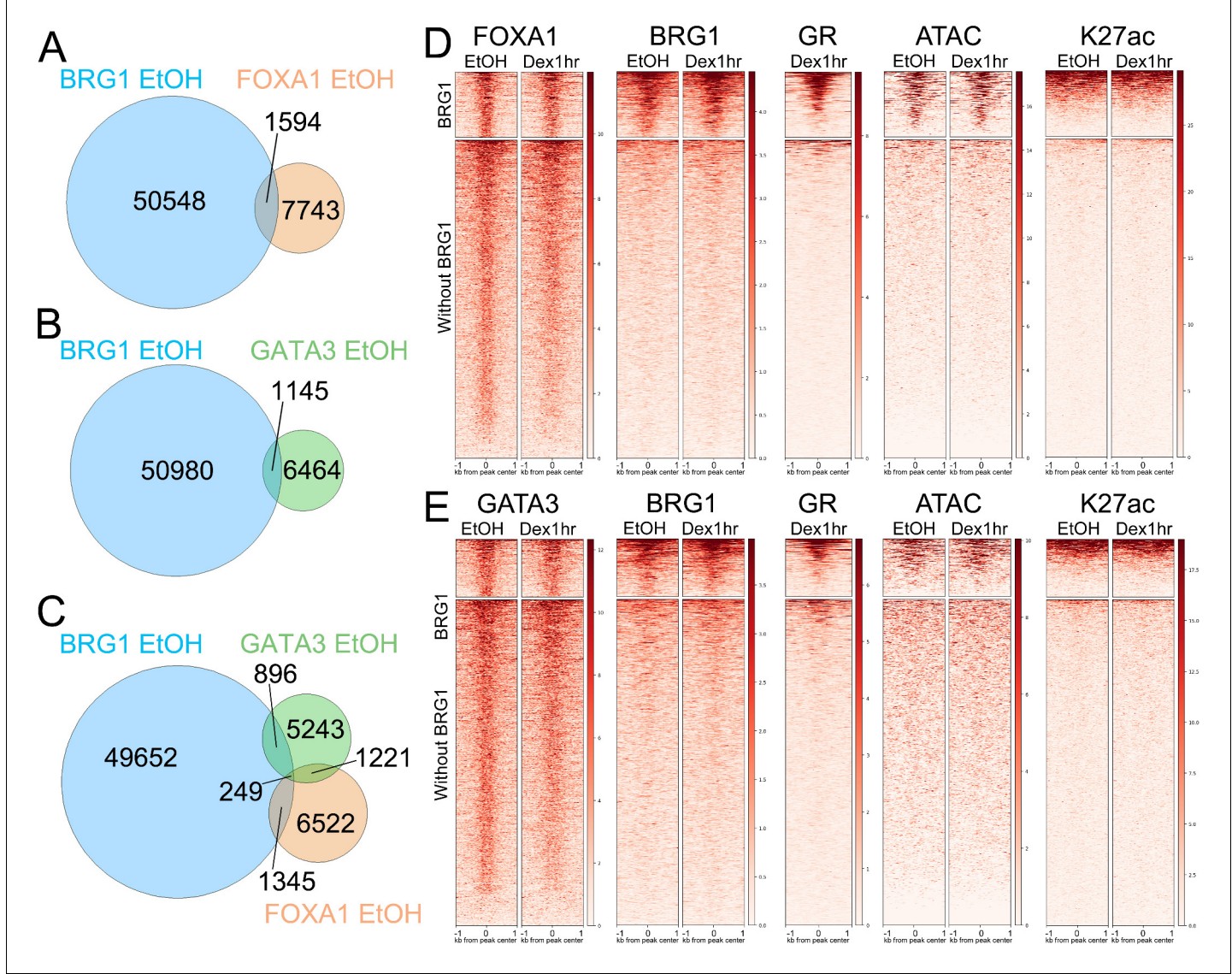

**Figure 6.** BRG1 binding to pioneer factor binding sites is predictive of GR binding upon Dex treatment. (**A**) Overlap between BRG1 EtOH and FOXA1 EtOH peaks. (**B**) Overlap between BRG1 EtOH and GATA3 EtOH peaks. (**C**) Three-way Venn diagram showing overlap of all three factors (**D**) Heatmaps depicting FOXA1, BRG1, GR, ATAC nucleosome free reads, and K27ac over FOXA1 EtOH peaks divided into 'with BRG1' and 'without BRG1' subsets. (**E**) Heatmaps depicting GATA3, BRG1, GR, ATAC nucleosome free reads, and K27ac over GATA3 EtOH peaks divided into 'with BRG1' and 'without BRG1' subsets. Note that in both (**D**) and (**E**) GR binding to FOXA1 and GATA3 peaks is largely restricted to 'with BRG1' peaks.

DOI: https://doi.org/10.7554/eLife.35073.016

The following figure supplement is available for figure 6:

**Figure supplement 1.** Proportion of GREs under pioneer factor peaks.

DOI: https://doi.org/10.7554/eLife.35073.017

associated with transcriptional activity, indicating that GR binding to chromatin is not sufficient to activate transcription. This is especially evident at Class I peaks, which are devoid of active chromatin markers and display minimal ATAC accessibility. Active and accessible chromatin was only detected at Class II GR peaks where BRG1 localization was constitutive/hormone-independent. This active chromatin environment was pre-existing, and was largely unchanged by Dex exposure and GR binding. However, half of Class II peaks do not have active TSSs within 1 kb, and not all Class II peaks have appreciable K27ac enrichment. Furthermore, BRG1 binding was also not sufficient to generate a fully activated chromatin environment at GR binding sites, as Dex-induced recruitment

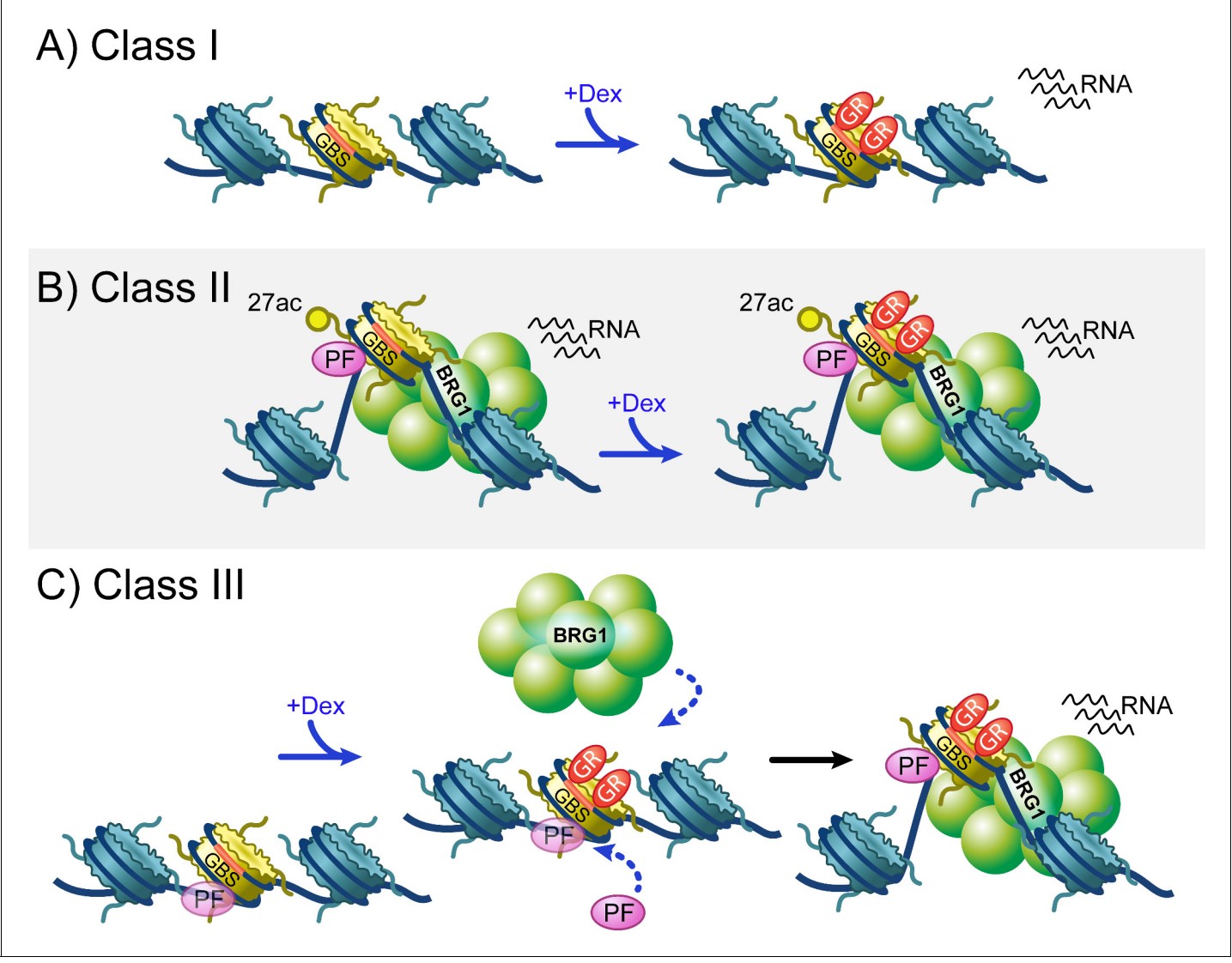

**Figure 7.** Overview of three classes of GR binding site. (**A**) Class I GR binding sites (GBSs) reside within relatively closed regions of chromatin and have little dex-dependent chromatin remodeling or recruitment of BRG1 and pioneer factors. Despite this, greater than a quarter of Class I GBSs exhibit dex-dependent transcription. (**B**) Class II GBSs represent GR binding to active and occupied regions of chromatin. These GBSs are bound by BRG1 prior to hormone treatment, and also exhibit hormone independent H3K27 acetylation, chromatin accessibility, pioneer factor binding, and transcriptional activity. (**C**) Class III GBSs behave the most like the model described at the MMTV promoter. Upon hormone treatment, GR binds to regions of relatively inaccessible chromatin that may be pre-occupied by low levels of pioneer factors. Upon GR binding, BRG1 and additional pioneer factors are recruited, chromatin remodeling yields increased accessibility, and more than a third of these GBSs gain transcriptional activity.
DOI: https://doi.org/10.7554/eLife.35073.018

of BRG1 at Class III peaks occurs without a concomitant induction of K27ac. The implications for these phenomena are wide-ranging. Enrichment of H3K4me1 at Class I and III GR peaks suggests that these binding sites are poised for transcriptional activation, and yet, Dex-induced binding of GR and BRG1 do not yield conversion of these sites to a more active chromatin profile such as that observed at Class II peaks. Thus, it appears that a large subset of GR chromatin interactions are transcriptionally unproductive and uneventful in terms of the effect on the chromatin environment. Despite this, Start-seq reveals that transcriptional activity is gained at ~25% of Class I peaks and ~30% of Class III peaks upon Dex exposure. This suggests that the induction of transcriptional activity at GR binding sites occurs independently of the induction of common active chromatin characteristics.

Intriguingly, ~60% of Dex-regulated DEGs are lost following BRG1 knockdown, and Class II and III GR binding sites represent ~60% of GR binding sites. The suppression of the GR transcriptional response following BRG1 knockdown suggested that BRG1 interaction with GR binding sites is required for GR-mediated transcriptional regulation of many genes. Surprisingly, over 100 genes gained hormone-responsiveness following BRG1 knockdown, indicating that at some genes, BRG1 prevents GR from eliciting transcriptional activity. Thus, BRG1 plays a critical role in patterning the GR transcriptional response. As the number of GR peaks is substantially greater than the number of GR-regulated genes, and most genes have multiple GR peaks of different classes within the surrounding several hundred kilobases, it is difficult to clearly associate individual GR peaks or peak classes with specific genes. The distal nature of GR binding events and the enhancer-like characteristics of the chromatin under GR binding sites indicate that the GR signaling largely regulates transcription through modulation of enhancer activity. On the other hand, BRG1 was enriched at most active gene and enhancer TSSs, implicating a widespread role for BRG1 in facilitating transcriptional activity. A reasonable hypothesis for GR signaling would be that BRG1 binding at gene TSSs and GR binding sites/enhancers promotes chromatin looping. Long-range chromatin interactions have been implicated in GR transcriptional activity (*Vockley et al., 2016*; *Hakim et al., 2009*). It has also been suggested that clusters of GBSs interact with each other over long ranges to synergistically regulate transcription of target genes *Holmqvist et al., 2005*; *Vockley et al., 2016*). Identifying such long-range interactions between GR and BRG1 would provide more insight into whether the different classes of GR binding sites are differentially utilized in regulating transcription. Such long range interactions could potentially provide functional rationale for the existence of 'unproductive' GR binding sites, which could be associated with transcriptionally active GR binding sites to cooperatively regulate transcriptional output. Alternatively, GR and BRG1 could regulate gene expression through decompaction of broad regions of chromatin surrounding gene TSSs and enhancers, such as has been reported at the Fkbp5 and Ms4xxx loci in macrophages (*Jubb et al., 2017*). In either case, removing individual or combinations of GR peaks around candidate DEGs will allow for interrogation of how multiple GR binding events are coordinated to elicit a transcriptional response. The eventual identification of which GR binding sites are critical for transcriptional regulation in a given cell type will have wide-ranging implications for the pharmacological targeting of GR signaling for disease treatment.

We observed that BRG1 binding at Class II binding sites is predictive of potential GR interactions with pioneer factor binding sites. This finding is intriguing when considered along with the recent finding that FOXA1 binding is reorganized upon activation of ER or GR (*Swinstead et al., 2016*). Like FOXA1, BRG1 binding is significantly reorganized upon hormone treatment and the majority of EtOH-specific and Dex-specific BRG1 peaks are not associated with GR binding sites [*Figure 1*, *Figure 1—figure supplement 1*]. Thus, the rearrangement of BRG1 and FOXA1 binding does not appear to be dependent on direct interaction with GR. While these rearrangements occur within an hour of hormone treatment, it is possible that the reorganization of BRG1 and FOXA1 binding occurs secondarily to the initial GR binding events, which occur rapidly within the first several minutes of hormone treatment (unpublished data). Alternatively, it is also possible that a subset of interactions between GR, BRG1 and FOXA1 on chromatin are not detectable by standard ChIP methods. Single molecule analysis of GR, BRG1, and FOXA1 indicated that the majority of chromatin interactions occur with residence times of approximately 1 s, and a minority of events occur with longer residence times of 5 to 10 s (*Swinstead et al., 2016*; *Paakinaho et al., 2017*). Thus, chromatin binding by these factors in individual cells is highly dynamic. While ChIP-seq experiments represent the overall binding profile of these factors in large populations of cells, they still represent snapshots of the chromatin interaction profiles of these factors and could fail to detect the full expanse of rapid and dynamic binding events. As such, it remains unclear whether there is any set hierarchy of binding or timing/order of events of FOXA1, BRG1, and GR chromatin interactions. As suggested by the distinct patterns of interactions at GR peak classes, it is likely that several series of binding events occur at GR binding sites prior to and upon hormone signaling. Elucidating these distinct mechanisms will help to unravel the basic mechanisms of nuclear receptor signaling and the role of pioneer factors and chromatin remodeling complexes in facilitating chromatin interactions and modulating transcriptional output.

# Materials and methods

## Key resources table

| Reagent type (species) or resource | Designation | Source or reference | Identifiers | Additional information |
|---|---|---|---|---|
| Cell line (*H.sapiens*) | A1-2 | PMID: 7838148 | RRID:CVCL_0I95 | T47D derivative with incorporated rat GR |
| Cell line (*H.sapiens*) | A1-A3 | PMID: 22451486 | | A1-2 derivative with incorporated BRG1 shRNA |
| Antibody | anti-BRG1 | PMID: 26055322 | | lab-made, ChIP = 1 ug/100 ug chromatin, Western Blot = 0.1 ug/ml |
| Antibody | anti-GR | Santa Cruz | M-20, sc-1004, RRID:AB_2155786 | ChIP = 1 ug/100 ug chromatin, Western Blot = 0.1 ug/ml |
| Antibody | anti-FOXA1 | Abcam | ab23738, RRID:AB_2104842 | ChIP = 1 ug/100 ug chromatin, Western Blot = 0.1 ug/ml |
| Antibody | anti-GATA3 | Cell Signaling | D13C9, RRID:AB_10834528 | ChIP = 1 ug/100 ug chromatin, Western Blot = 0.1 ug/ml |
| Antibody | anti-H3K27ac | Abcam | ab4729, RRID:AB_2118291 | ChIP = 1 ug/100 ug chromatin |
| Antibody | anti-H3K27me3 | Active Motif | 39155, RRID:AB_2561020 | ChIP = 1 ug/100 ug chromatin |
| Antibody | anti-H3K4me1 | Abcam | ab8895, RRID:AB_306847 | ChIP = 1 ug/100 ug chromatin |
| Antibody | anti-KU70 | Santa Cruz | H-308, sc-9033, RRID:AB_650476 | Western Blot = 0.1 ug/ml |
| Sequence-based reagent | GR siRNA | Dharmacon | ON-TARGETplus J-089504–07 | UUACAAAGAUUGCAGGUAU |
| Sequence-based reagent | Non-targeting Control siRNA | Dharmacon | ON-TARGETplus Not-targeting Pool D-001810-10-20 | |
| Commercial assay or kit | Nextera XT library generation kit | Illumina | 15032350 | |
| Commercial assay or kit | SuperScript III First-Strand kit | Invitrogen | 18080–051 | |
| Commercial assay or kit | iScript cDNA Sythesis kit | Bio-Rad | 170–8891 | |
| Commercial assay or kit | ssoAdvanced Universal SYBR Green Supermix | Bio-Rad | 172–5274 | |
| Commercial assay or kit | RNeasy Mini Kit | Qiagen | 74104 | |
| Commercial assay or kit | RNA 6000 RNA Pico Kit | Agilent Technologies | 5067–1513 | |
| Commercial assay or kit | QiaQuick PCR purification kit | Qiagen | 28104 | |
| Commercial assay or kit | HALT protease inhibitors | ThermoFisher | 78430 | |
| Chemical compound, drug | Dexamethasone | Sigma | D4902 | 100 nM |
| Chemical compound, drug | Doxycycline | Sigma | D9891 | 10 ug/ml |
| Software, algorithm | Cutadapt | DOI: http://dx.doi.org/10.14806/ej.17.1.200 | RRID:SCR_011841 | |
| Software, algorithm | Sickle | https://github.com/najoshi/sickle | RRID:SCR_006800 | |

*Continued on next page*

*Continued*

| Reagent type (species) or resource | Designation | Source or reference | Identifiers | Additional information |
|---|---|---|---|---|
| Software, algorithm | Bowtie2 | PMID: 22388286 | RRID:SCR_005476 | |
| Software, algorithm | Samtools | PMID: 19505943 | RRID:SCR_002105 | |
| Software, algorithm | MACS2 | PMID: 18798982 | RRID:SCR_013291 | |
| Software, algorithm | Homer | PMID: 20513432 | RRID:SCR_010881 | |
| Software, algorithm | Bedtools | PMID: 20110278 | RRID:SCR_006646 | |
| Software, algorithm | Deeptools | PMID: 27079975 | | |
| Software, algorithm | AME | DOI: https://doi.org/10.1186/1471-2105-11-165 | RRID:SCR_001783 | http://meme-suite.org/tools/ame |
| Software, algorithm | STAR | PMID: 23104886 | RRID:SCR_015899 | |
| Software, algorithm | Salmon | PMID: 28263959 | | |
| Software, algorithm | limma-voom | PMID: 24485249 | RRID:SCR_010943 | |

## Cell culture

T47D derived A1-2 (*Archer et al., 1994*) and A1-A3 (*Burd et al., 2012*) cells were cultured as previously described (*Burd et al., 2012*). Both cell lines were authenticated by STR profiling and tested negative for mycoplasma. Dexamethasone treatments were performed using 100 nM Dexamethasone or ethanol vehicle for 1 or 8 hr for ChIP-seq and RNA-seq experiments, respectively. To knock-down BRG1 expression in A1-A3 cells, cells were treated for 72 hr with Doxycycline.

## ChIP and ChIP-seq

ChIP experiments were largely performed as previously described (*Takaku et al., 2016*). Cells were fixed with 1% formaldehyde at 37C for 10 min for all targets except BRG1, for which cells were fixed for 20 min. After quenching with glycine, cell pellets were washed Hypotonic buffer (10 mM HEPES-NaOH pH 7.9, 10 mM KCl, 1.5 mM MgCl2, 340 mM sucrose, 10% glycerol, 0.1% Triton X-100, and HALT protease inhibitors (ThermoFisher)) and resuspended in Shearing buffer (10 mM Tris-HCl pH 8.0, 1 mM EDTA, 0.5 mM EGTA, 0.5 mM PMSF, 5 mM Sodium Butyrate, 0.1% SDS, and HALT protease inhibitors (ThermoFisher) and chromatin was fragmented by sonication with the Covaris S220. Chromatin was diluted two-fold in 2xIP buffer (20 mM Tris-HCl pH 8.0, 300 mM NaCl, 2 mM EDTA, 20% Glycerol, 1% Triton X-100, 0.5 mM PMSF, 5 mM Sodium Butyrate, and HALT protease inhibitors (ThermoFisher)) and immunoprecipitation was performed with antibodies specific to BRG1 (lab-made, [*Trotter et al., 2015*]), GR (Santa Cruz M-20), FOXA1 (Abcam ab23738), GATA3 (Cell Signaling D13C9), and H3K27ac (Abcam ab4729) and ratios of 1 ug antibody per 100 ug chromatin. Immune complexes were captured using protein A and G dynabeads, washed once each with low salt (20 mM Tris-HCl pH 8.0, 150 mM NaCl, 2 mM EDTA, 1% Triton X-100, 0.1% SDS), high salt (same as low salt buffer, except 500 mM NaCl), and LiCl buffer (Tris-HCl pH 8.0, 250 mM LiCl, 2 mM EDTA, 1 % NP-40, 1% (wt/vol) sodium deoxycholate), and twice with TE. Eluted DNA was RNaseA and Proteinase K treated and purified using Qiagen PCR purification columns. ChIP-seq libraries were generated using the Illumina Nextara-XT library generation kit, and sequenced on the Illumina MiSeq and NextSeq platforms. For all ChIP-seq experiments, biological duplicates or triplicates were performed, and all presented ChIP-seq data are representative single experimental replicates. Examples of reproducibility of multiple replicates are presented in *Figure 1—figure supplement 2*.

Adapter sequences were trimmed from ChIP-seq reads using Cutadapt (*Martin, 2011*) and low quality reads were removed from analysis using Sickle (*Joshi NA et al., 2011*). Alignment was performed with Bowtie2 (*Langmead and Salzberg, 2012*). Aligned reads were sorted and processed with Samtools (*Li et al., 2009*) and de-duplicated using Picard Tools (http://broadinstitute.github.io/picard). Peaks were called using MACS2 (*Zhang et al., 2008*) and Homer (*Heinz et al., 2010*) using a false discovery rate cutoff of 0.001, and regions of high depth or with high signal in untreated or input samples were used to filter out false positive peak calls. Peak overlaps and distance analyses were performed using Bedtools (*Quinlan and Hall, 2010*). Coverage files and heatmaps were

generated using Deeptools (*Ramírez et al., 2016*). Motif analyses were performed using AME (*McLeay and Bailey, 2010*).

## RNA isolation, cDNA synthesis, QPCR, and RNA-seq

RNA was isolated from treated A1-2 and A1A3 cells using Qiagen RNeasy kits with on-column DNase treatment. ThermoFisher SuperScript III or BioRad iScript were used to synthesize DNA and qPCR was run with BioRad ssoAdvanced Universal SYBR Green Supermix. For RNA-seq, RNA quality was validated with RNA 6000 RNA Pico Kit on the Agilent Bioanalyzer 2100. RNA-seq libraries were generated at the National Intramural Sequencing Center using Ribo-Zero Gold and sequenced on an Illumina HiSeq 2500. Adapter sequences were trimmed from RNA-seq reads using Cutadapt (*Martin, 2011*) and low quality reads were removed from analysis using Sickle (*Joshi NA et al., 2011*). Alignment was performed using STAR (*Dobin et al., 2013*) to generate coverage tracks and using Salmon (*Patro et al., 2017*) and to obtain gene counts for differential expression analysis using limma-voom (*Law et al., 2014*) with cutoffs of Fold Change > 1.5, p-value<0.01, and False Discovery Rate < 0.05.

## Data availability

All ChIP-seq and RNA-seq data generated for this publication have been deposited in NCBI's Gene Expression Omnibus (*Edgar et al., 2002*) and are accessible through GEO Series accession number GSE112491 (https://www.ncbi.nlm.nih.gov/geo/query/acc.cgi?acc=GSE112491).

## Acknowledgements

We thank members of the Archer group, the NIEHS Epigenetics and Stem Cell Biology Laboratory, and the NIEHS Integrative Bioinformatics Group for their ongoing support, advice, and constructive criticism. We also thank Greg Solomon and Jason Malphurs of the NIEHS Epigenomics Core Laboratory for their next generation sequencing expertise. Finally, we are grateful to Paul Wade and David Fargo for critical review of the manuscript.

## Additional information

### Funding

| Funder | Grant reference number | Author |
| --- | --- | --- |
| National Institute of Environmental Health Sciences | Z01 ES071006-17 | Trevor K Archer |

The funders had no role in study design, data collection and interpretation, or the decision to submit the work for publication.

### Author contributions

Jackson A Hoffman, Conceptualization, Data curation, Formal analysis, Validation, Investigation, Visualization, Methodology, Writing—original draft, Writing—review and editing; Kevin W Trotter, Conceptualization, Validation, Investigation, Writing—review and editing; James M Ward, Data curation, Formal analysis, Visualization; Trevor K Archer, Conceptualization, Resources, Supervision, Funding acquisition, Validation, Project administration, Writing—review and editing

### Author ORCIDs

Jackson A Hoffman http://orcid.org/0000-0002-6256-144X
Trevor K Archer http://orcid.org/0000-0001-7651-3644

### Decision letter and Author response

Decision letter https://doi.org/10.7554/eLife.35073.023
Author response https://doi.org/10.7554/eLife.35073.024

## Additional files

### Supplementary files

• Transparent reporting form
DOI: https://doi.org/10.7554/eLife.35073.019

### Data availability

Sequencing data have been deposited in GEO under accession code GSE112491.

The following dataset was generated:

| Author(s) | Year | Dataset title | Dataset URL | Database, license, and accessibility information |
|---|---|---|---|---|
| Jackson A Hoffman, Kevin W Trotter, James M Ward, Trevor K Archer | 2018 | BRG1 governs Glucocorticoid Receptor interactions with chromatin and pioneer factors across the genome | https://www.ncbi.nlm.nih.gov/geo/query/acc.cgi?acc=GSE112491 | Publicly available at the NCBI Gene Expression Omnibus (accession no: GSE112491) |

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
