## [Decision Letter]

Thank you for submitting your article "BRG1 defines distinct chromatin states and pioneer factor utilization at nuclear receptor binding sites" for consideration by *eLife*. Your article has been reviewed by three peer reviewers, and the evaluation has been overseen by a Jerry Workman as Reviewing Editor and Jessica Tyler as the Senior Editor. The reviewers have opted to remain anonymous.

The reviewers have discussed the reviews with one another and the Reviewing Editor has drafted this decision to help you prepare a revised submission.

Summary: It is important to understand the mechanism by which glucocorticoid hormone regulates transcription, because several clinically important drugs, including dexamethasone (Dex), act through this pathway. Chromatin structure plays a critical role in determining which sites are bound by the hormone-activated glucocorticoid receptor (GR). The authors describe a genomic study of the role of the BRG1 chromatin remodeler in Dex-induced binding of GR and transcription activation in a human breast cancer cell line. They show that the presence or absence of BRG1 can distinguish three specific classes of GR binding site chromatin. One of these classes (Class II) corresponds primarily to GR-site-containing enhancers that are already active before hormone addition; they are already associated with BRG1, the H3-K27ac active mark, the FOXA1 and GATA3 pioneer factors, and the enhancer DNA is accessible. Addition of Dex results in relatively strong binding of GR and increased binding of FOXA1 and GATA3 pioneer factors. They show that knock down of BRG1 prevents Dex-dependent binding of these pioneer factors. Of the other classes of GR site, Class I sites are not associated with BRG1 at all, indicating that GR binding is independent of BRG1, whereas Class III sites are associated with BRG1 only after Dex treatment. Both of these classes of site are associated with H3-K4me1 (which marks inactive/poised enhancers) and are less accessible than Class II enhancers. Overall, the authors provide important new insight into the complexity of GR binding site selection, the role of the BRG1 remodeler and the link to pioneer factors. This study is thorough, describes a lot of new data, and the paper is nicely written.

Essential Revisions:

1) A primary concern is that BRG1 knockdown (only 50% effective on WB) leads to ~50% decrease in GR nuclear translocation (Figure 3—figure supplement 1A), which was ignored by the authors and could per se explain the changes in GR binding and hormone responsiveness observed at the genomic level. Can the authors experimentally rule out the possibility that the effects they attribute to reduced BRG1 are in fact due to reduced GR? While it may be difficult to make the clean separation of nuclei from cytoplasm that is necessary to distinguish between nuclear Dex-bound activated GR and cytoplasmic inactive GR, control markers for the WB for nuclear and cytoplasmic markers may help.

2) The number of peaks that the authors identify is high (GR binding to 20K^+^ sites) with relative low functional significance as only ~1200 differentially expressed genes (DEGs) were identified. The authors use MACS2 to call peaks and they seem to have a common problem with this type of analysis (over-calling peaks). They used regions of high depth in the untreated and input samples to filter out false positives but the discrepancy between the high number of GR-binding sites and the vastly lower number of functionally relevant DEGs brings the effectiveness of this method in question.

3) Another issue is that the authors used 8 hours of Dex treatment in the transcription studies (Figure 3), but the rest of the data are for 1 hour of Dex. The authors should comment on whether the difference in Dex treatment times could be important.

4) The authors should consider adding a "cartoon" figure as a summary of the properties of the three classes of GR site.

---

## [Author Response]

Essential Revisions:

1) A primary concern is that BRG1 knockdown (only 50% effective on WB) leads to ~50% decrease in GR nuclear translocation (Figure 3—figure supplement 1A), which was ignored by the authors and could per se explain the changes in GR binding and hormone responsiveness observed at the genomic level. Can the authors experimentally rule out the possibility that the effects they attribute to reduced BRG1 are in fact due to reduced GR? While it may be difficult to make the clean separation of nuclei from cytoplasm that is necessary to distinguish between nuclear Dex-bound activated GR and cytoplasmic inactive GR, control markers for the WB for nuclear and cytoplasmic markers may help.

This is an excellent point, and we agree that we did not sufficiently address the apparent decrease in GR protein levels in BRG1 knockdown conditions. To address this directly, we have performed a new experiment to reduce GR protein levels by siRNA and compare the effects on target gene expression with those observed in the BRG1-KD conditions. Briefly, while GR knockdown did reduce the dex-induction of “lost” target genes, the effects caused by BRG1 knockdown were greater. This suggests that, while the reduction of GR levels following BRG1-KD may have some effect on transcription, knockdown of BRG1 is major driver of this altered gene expression.

We have added two new panels to Figure 3—figure supplement 1 and added an additional paragraph discussing this to the Results section (subsection “BRG1 is required for the transcriptional hormone response”, third paragraph.

2) The number of peaks that the authors identify is high (GR binding to 20K^+^ sites) with relative low functional significance as only ~1200 differentially expressed genes (DEGs) were identified. The authors use MACS2 to call peaks and they seem to have a common problem with this type of analysis (over-calling peaks). They used regions of high depth in the untreated and input samples to filter out false positives but the discrepancy between the high number of GR-binding sites and the vastly lower number of functionally relevant DEGs brings the effectiveness of this method in question.

The reviewers raise an excellent point here that we have also raised in our Discussion section. We are confident in our GR peak calls for several reasons:

1) Our GR, BRG1, and input ChIP-seq experiments were sequenced at high-depth (greater than 60 million reads each) and produced high quality data with low signal-to-noise.

2) We called peaks using a fairly stringent 0.001 False Discovery Rate cutoff, and we have added this information to the Materials and methods section of the manuscript.

3) Calling peaks with Homer and using the same 0.001 False Discovery Rate threshold resulted even more peaks being called (~58,700).

4) Previously published data sets have identified a wide range of GR peaks in different cells types, including data sets in IMR90 and U2OS cells which identified >40,000 GR peaks in each cell line (Starick et al., 2015). We have added a sentence indicating this to the Results section.

5) It has been proposed that clusters of multiple GR binding sites interact and synergize to regulate transcriptional activity (Vockley CM et al., Cell 2016). We have added a reference to this finding in our Discussion section (third paragraph).

Thus, abundance of GR peaks relative to DEGs is an intriguing feature of GR transcriptional regulation and we propose further investigation of this phenomenon in our Discussion section.

3) Another issue is that the authors used 8 hours of Dex treatment in the transcription studies (Figure 3), but the rest of the data are for 1 hour of Dex. The authors should comment on whether the difference in Dex treatment times could be important.

We agree that we did not sufficiently present the reasoning behind our choice of the 8hr Dex treatment timepoint for our A1A3 RNAseq experiment. We reasoned that 1hr treatment could potentially be insufficient to capture a “full” transcriptional response to Dex, and that an 8hr treatment would provide a greater panel of DEGs. Indeed, at 8hrs Dex we identified ~5x more DEGs than previously detected at 1hr by RNAseq in A1-2 cells. We have added an appropriate description of our rationale to the Results section.

4) The authors should consider adding a "cartoon" figure as a summary of the properties of the three classes of GR site.

We thank the reviewers for this suggestion. We have added a cartoon summary of the three peak classes as Figure 7 and referenced this figure in the Discussion section (first paragraph).